# Microbial Growth: Role of Water Activity and Viscoelasticity of the Cell Compartments

**DOI:** 10.3390/ijms26178508

**Published:** 2025-09-01

**Authors:** Alberto Schiraldi

**Affiliations:** Formerly at Department of Food Environmental Nutrition Sciences (DeFENS), University of Milan, 20133 Milan, Italy; alberto.schiraldi@unimi.it

**Keywords:** no-growth limits, microbial viability, water activity, glass transition temperature, visco-elastic properties

## Abstract

The complexity of the biochemistry and the variety of possible environments make the subject of the no-growth limits of bacteria a very tough challenge. This present work addresses the problem of applying to the microbial cultures the polymer science approach, which is widespread in food technology. This requires the definition of a “dynamic state diagram” that reports the expected trends of the glass transition of two virtual polymers, which mimic the crowded cytoplasmic polymers and the polymeric meshwork of the cell envelope, respectively, versus the water content. At any given temperature, the water content at the glass transition represents the lowest limit for the relevant molecular mobility. This representation leads one to recognize that the lowest temperature to observe microbial growth coincides with that of the largest freeze-concentrated liquid phase, in line with the values predicted by the Ratkowsky empirical equation. In view of potential applications in predictive microbiology, this paper suggests an alternative interpretation for the highest tolerated temperature and the temperature of the largest specific growth rate.

## 1. Introduction

Liquid water is ubiquitous in biological organisms and related food, pharmaceutical, and cosmetic products, where it substantially affects the overall physical properties (density, viscoelasticity, rheology, etc.), the chemical stability (oxidation, enzymatic, and non-enzymatic deterioration, etc.), and the susceptibility to microbial spoilage.

For decades, relative humidity, *RH*, and temperature, *T*, seemed to be the main physical parameters to control the detrimental effects related to moisture in food and pharmaceutical products: drying, freeze-drying, osmo-dehydration, cooling, salting, etc., became usual technological preservation practices. The relevant industrial implementation required quantitative estimation of the optimal *RH* and *T* levels for each product, or kind of product, as well as of the lowest *RH* and *T* compatible with the growth of a number of microbes [1,2,3,4,5,6], including viruses [7].

The “Labuza Map” (from the name of the pioneer in this field) reports the observed correlation between *RH* and the rate of many detrimental processes, including the microbial growth, at room temperature [2]. As for the effects of temperature, the so-called “Polymer Science Approach” by Slade and Levine [8] highlighted the role of the glass transition temperature, *T*_g_, and suggested the use of a dynamical state diagram in the (*T*, *C*_W_) plane to define the boundaries of phases (ice, liquid, glassy/amorphous) for a water/polymer binary [9,10]. A substantial improvement of this approach came with the concept of “critical water activity”, *a*_WC_, which is the result of coupling the *T*_g_-vs.-*C*_W_ trend with the sorption isotherm [3]. In line with this view, Rahman [11] suggested enhanced stability criteria for food systems, associating the role of *a*_W_ with the distance (because of either *T* or *C*_W_) from the glassy/rubbery state, where the molecular mobility is very poor.

On the microbiological side, Ratkowsky [12] considered the effects of *T* on the microbial growth. He observed that the square root of the maximum specific growth rate, *μ*, of most microbial cultures shows a straight-line dependence on *T* that allows the easy identification of *T*_min_ as the extrapolated temperature for which *μ* = 0. This approach (and its enhancements; see below) is deliberately empirical and has no connection with the polymer science approach and the stability criteria mentioned above.

This present paper suggests a polymer science approach for the microbial cultures, as long as the cytoplasm and external envelope of bacterial cells show viscoelastic properties similar to those of glass-forming colloids and hydrated polymers, respectively. These properties show a correlation with the water activity of the surrounding medium. The first objective is a *T*_g_-vs.-*C*_W_ dynamic state diagram, where one can identify a viability region for microbial cells. Once coupled with the sorption isotherms of biopolymers that mimic the behavior of cytoplasm and cell envelope, this perspective becomes suitable to suggest a physical meaning for the no-growth *a*_W_ and *T*_min_ limits and justifies why both properties show values in similar ranges, namely, 0.99–1 and 260–280 K, respectively, for different kinds of microbes [2,7].

## 2. Viability of the Cytoplasm for a Closed Batch Microbial Culture

A viable microbial cytoplasm is not a system in thermodynamic equilibrium, since all the cytoplasmic biochemical processes require the fast displacement of small molecules, which have to interact with macromolecules (enzymes, nucleic acids, layered lipo- and gluco-proteins, polysaccharides, etc.) at specific active sites. Translational and/or segmental molecular mobility of cytoplasmic polymers within such a crowded ambient is a physical prerequisite for the viability of the cell [13,14,15,16]. If their mobility is naught, the biochemical machinery becomes shaky or inactive.

Water plays the role of “lubricant”, favoring snaking movements of the polymer chains through segmental adjustments, thanks to the disruption and reconstruction of intra- and intermolecular hydrogen bonds [17]. This is why large amounts of intracellular water “disappear”, being entrapped within the cytoplasmic polymeric entanglements that hinder easy exchanges with the surrounding bulk water, which represents a small fraction of the water content and, for polymer concentrations beyond 40% vol, experiences the electric field of the solute [14,18]. This means that the molecules of bulk water too are relented, which corresponds to a large relative viscosity, *η*_r_, of the cytoplasm.

Because of the easy transfer of water through the cell wall, one could expect the medium and cytoplasm to have the same *a*_W_, thus implying an “osmotic” equilibrium between cells and their immediate surroundings [19]. However, this is not the case, as the microbial culture (cells + medium) is rather far from the thermodynamic equilibrium, the rates of the intracellular reactions being diffusion limited [18,20] because of the reduced mobility of both large and small molecules.

One has to look at this system from a dynamic perspective. The cell metabolism implies the transformation of SMMS (small molecular mass substrates) into new biopolymers, which will fix large amounts of water (up to ten times their own mass). This produces an extra uptake of water, mainly from the outer medium. This water flow from the medium to the cytoplasm does not correspond to a true osmotic effect but is the result of the “sequestering” action by the heavy and poorly mobile biopolymers [20,21] involved in the biochemical activity of the viable cytoplasm. The changes in the intracellular world require sophisticated experimental approaches [22,23], but the description of the extracellular aqueous medium, fortunately, is much simpler, as relatively small changes in the medium mainly concern SMMS (catabolites replace substrate, pH can vary, while some water migrates into the cells) and are easily detectable by the investigator. The difficulty to face, therefore, is to infer correct prediction of intracellular events from the reliable information about the extracellular medium.

It can be of help to approach the proposed in previous papers [24,25] that suggest a formal correlation between thermodynamic and rheological properties to account for water exchanges between a homogeneous liquid phase and a phase that shows a viscoelastic and, possibly, non-isotropic behavior. This approach treats the system with a large relative viscosity, *η*_r_, by assigning extra terms to the chemical potential of water,(1)$$\mu_{W}=\left[\mu_{W}^{*}+RTln{\left(\eta_{r}\right)}^{\alpha }-RTln(H)\right]+RTln\left(a_{W}\right)$$
where *α* ≥ 0 (for solutions of small molecules, *α* ≈ 0) and *H* accounts for the heterogeneity-related hurdles (e.g., membranes, cell walls, etc.) and has a value that depends on the direction (inbound or outbound) of the water flow.

Equation (1) predicts that the condition for no net flow of water, namely a steady state (still not a true thermodynamic equilibrium) between medium (1), where *α*_1_ ≈ 0, and cytoplasm (2) would be(2)$$a_{W,1}=a_{W,2}\;{\left(\eta_{r,2}\right)}^{\alpha_{2}}\frac{H_{2,1}}{H_{1,2}}\;\approx \;a_{W,2}\;{\left(\eta_{r,2}\right)}^{\alpha_{2}}$$

Now, since *α*_2_ > 0 and *η*_r,2_ > 1, one may conclude that *a*_W,1_ > *a*_W,2_. In other words, water activity is smaller in the cytoplasm than in the outer medium, which seems consistent with the slightly lower freezing point of the cytoplasm [4,22]. It is worth noticing that this conclusion concerns just the bulk water of the cytoplasm. The water fraction sequestered by the cytoplasmic polymers does not directly contribute to the colligative properties of the cytoplasm and substantially corresponds to the so-called “unfreezable” water.

The balance between bulk and bound water fractions (which is around 50% of the overall water content of the cytoplasm, depending on the temperature and the kind of cell) seems to be the result of self-controlled feedback by the viable cells. Such feedback keeps (or quickly restores) a constant *a*_W,2_ [26], normally at the expense of the water content of the surrounding medium, whose water activity, *a*_W,1_, decreases.

It is now possible to envisage an adverse condition that compromises the viability of the cytoplasm at rest (at the temperature of the culture), namely,(3)$$a_{W,1}=a_{WC}\leq \;a_{W,2}\;{\left(\eta_{r,2}\right)}^{\alpha_{2}}$$

The cytoplasm would tend to release instead of uptake water with an increase in *η_r_*_,2_, while the feedback action mentioned above would remove water molecules from some cytoplasmic macromolecules, making the polymeric entanglement less plastic, looking like a colloidal glass. Such *a*_WC_ is the critical water activity (of the medium) at which the cytoplasmic polymers become too rigid to sustain the cell viability [13].

The metabolic activity of the cell ends when the flow of water ceases, namely, when the system attains the condition described by Equation (3), which therefore defines the lowest *a*_W_ compatible with the cytoplasmic viability.

## 3. No-Growth Water Activity

Viability of cytoplasm is a necessary but not sufficient condition for cell duplication. Microbial cells can indeed survive and even produce catabolites when immobilized and/or unable to duplicate because of an external mechanical constraint [27]. A natural constraint is the stiffness of the cell wall that allows the cell to keep its shape and bear osmotic stresses up to several atmospheres. In view of the cell duplication, one has to account also for this constraint.

Vast recent literature covers the theme of the cell wall of either Gram+ or Gram− bacteria [27,28,29,30,31,32,33]. The former have an external shield of murein, which is a reticulated meshwork of peptidoglycan (PG) chains, while the latter are surrounded by an inner layer of phospholipids (PL) and an outer layer of lipopolysaccharides (LPS) with an intermediate aqueous phase (periplasm) crossed by PG thin mesh. The rigidity of either kind of shield depends on the underlying cytoplasmic biochemistry [32,33,34] that in turn depends on the molecular mobility of aqueous cytoplasmic polymers, which therefore governs also the flexibility of the cell wall.

The PG cortex behaves like a viscoelastic meshwork [27,28,29,30,31,32,33] with variable rigidity, reflected by the relevant elastic modulus, *E*’. It must become plastic, thanks to the selective action of hydrolases (autolysins), when the cell undergoes duplication, while it is rigid during the preceding metabolic steps and in the newborn cell, allowing water and SMMS to trespass the meshwork barrier through pores and channels. The enzymes that govern the cleavage and reconstruction of the PG meshwork come from the cytoplasmic side, as well as the aquaporins that sustain the fast *a*_W_ equilibration between the hydration shell of the PG moieties, the aqueous medium, and the cytoplasmic bulk water.

In the case of Gram− bacteria, the rigidity of the outer shield mainly depends on the very tightly packed layer of the LPS bridged by Mg^2+^ and Ca^2+^ cations and changes depending on the ratio between saturated and unsaturated lipids, which is under the control of devoted cytoplasmic enzymes [34].

The focus of interest of the present work deals with the correlation between *a*_W_ and viscoelastic properties of the cell polymeric components, namely, the molecular mobility of cytoplasmic macromolecules and the flexibility of the outer shield.

It is worth remembering that such correlation between water activity and rheological properties has no thermodynamic foundation [25]. However, it allows the definition of the meaningful threshold of the “critical water activity”, *a*_WC_, which, in the present case, is the value of *a*_W_ of the culture medium that corresponds to the transition from large to small molecular mobility, namely, from small to large *E*’ of the outer shield and *η*_r_ of the cytoplasm.

In polymer science, this transition, commonly dubbed as “glass transition” or “glass-rubbery transition” actually encompasses a temperature range around the so-called glass transition temperature, *T*_g_. In the case of aqueous polymers, this transition depends on the water mass fraction, *C*_W_ = *M*_W_/*M*.

The so-called “dynamic state diagram” [35] reports in the plane (*T*, *C*_W_) the phases (liquid, ice, glassy/amorphous polymer) of an aqueous binary system. Such a diagram allows a clear view of the regions within which one may expect a large molecular mobility and then the viability of the microbial cells.

In the case of food products, the practical use of the dynamic phase diagram for systems that host a number of different polymers requires reference to a single virtual polymer with analogous rheological properties, namely, the macroscopic evidence of the *T*_g_ trend determined experimentally. To this scope, suitable experimental approaches are calorimetric (DSC) or dinamico-mechanical (DMA) investigations on samples with different *C*_W_ or *a*_W_) hosted in sealed crucibles [36,37,38,39].

In the case of microbial cultures, the necessary information requires techniques that allow one to single out the glass transition of the cell interior from that of its rigid envelope. The experimental evidence must indeed concern the properties of the single cells. This makes unsuitable the results collected (with various methods) from “slurries” of cells (samples of some milligrams) [40], which therefore deal with collective properties. These are important to understand how a crowd of microbes tends to occupy the smallest volume or to form films over supporting surfaces or to investigate the survival in very low *a*_W_ environments, but are useless to sketch a dynamic phase diagram.

The analysis of the incoherent elastic neutron scattering is the method proposed by Sogabe et al. [41], who related the mean square displacement (MSD) of hydrogen atoms to the molecular mobility within *Cronobacter sakazakii* cells. Live cell fluorescence microscopy [42] is another promising approach but still has to provide viscoelasticity data. The CLAMP (Cell Length Analysis of Mechanical Properties) technique allows determination of the elastic modulus of the cell envelope by observing the elongation of living cells entrapped within agarose gels of different stiffness [27], but does provide estimation of *a*_WC_, or *T*_g_.

The best pieces of information come from Atomic Force Microscopy (AFM) approaches, which use a nanometric tip mounted at the end of a cantilever that mechanically solicits the cell wall and allows collection of stress relaxation data [43,44]. By applying different stresses and/or imposing different strains at various frequencies, one can determine the complex modulus, *E**, and its components, *E*’ and *E*”. In simple words, gentle stresses (about 1 N) and small strains (less than 10 nm indentation of the tip) reveal the viscoelasticity of the rigid component of the cell, namely, its outer envelope, which shows the behavior of an elastic body (*E*’ >> *E*”). For larger stresses and deeper indentations of the tip, the *E*” component tends to prevail, as in the case of viscous fluids, just what is expected from the cytoplasm.

It turns out that the external envelopes of either Gram− or Gram+ bacteria, in spite of their chemical differences, have elastic moduli of the same order of magnitude (and are much larger than for mammalian cells) [27,45]. This supports the hypothesis that the outer membrane of Gram-negative bacteria can contribute to the overall mechanical rigidity of their envelope [34]. This means that, in the dynamic phase diagram of cultures of either Gram+ or Gram− bacteria, a single virtual polymer can represent the cell envelope, while a different virtual polymer is necessary to reflect the mechanical properties of the cytoplasm that is much more plastic than the cell envelope.

Unfortunately, the available data do not allow a quantitative definition of a *T*_g_-vs.-*C*_W_ trend. However, the widely collected evidence shows that a rigid envelope coexists with a soft cytoplasm. This means that, at the temperature of the microbial culture (say, 298 K), the former and the latter are below and above their own *T*_g_, respectively. This allows the sketch of the expected trends (Figure 1). A single virtual polymer, P, mimics the average rheological properties of the cytoplasmic polymeric crowd, while another virtual polymer, PG, reflects the properties of the outer shield. Along a tie line at constant *T*, lowering *C*_W_ corresponds to increasing molecular crowding (or drying) and reduced molecular mobility, while the opposite occurs when increasing *C*_W_.

For a given *C*_W_, the glass transition trend of PG, *T*_g,PG_, runs above the *T*_g,P_ trend, i.e., *T*_g,PG_ > *T*_g,P_. The expected slope of either *T*_g_, for the driest conditions, is about 10 degrees Celsius per 1% increase in water mass ratio [8,9,10]. The region of the diagram between the *T*_g,PG_ and *T*_g,P_ trends corresponds to the condition of “viable” cytoplasm, as the relevant polymers acquire some mobility, being capable of short-range displacements [46,47] within a rigid capsule that does not allow cell duplication because of the mechanical constraint. The state of non-duplicating, but still viable, cells would correspond to a plastic gel entrapped in a rigid envelope, where some activity can still take place, since mobile SMMS can reach the almost immobilized endo-cellular enzymes. The cytoplasm behaves as a glass-forming liquid approaching the relevant glass transition. Its biopolymers lose their translation degrees of freedom, which damps the cell biochemical machinery [1,9,13].

This region of the dynamic state diagram encompasses the *C*_W_ range smaller than the maximal freeze concentration, *C*_g_’ (the cross point between the glass transition trend of the polymer and the decreasing trend of the cryoscopic ice separation [35]). The corresponding water, therefore, is “unfreezable”, playing just the role of conditioner of the viscoelastic properties of the cytoplasmic and wall polymers.

The *C*_W_ > *C*_g_’ region of the diagram reflects the properties of the aqueous solution of SMMS, namely, it undergoes the phase changes expected on varying the temperature, like the decreasing trend of the ice separation and the increase in boiling point.

The region of the diagram above the *T*_g,PG_ trend corresponds to the condition of more mobile cytoplasmic polymers within a flexible and modifiable outer envelope: cell duplication may take place.

The dashed area of the diagram in Figure 1 therefore represents the “Life Zone”, while the region below the *T*_g,P_ trend corresponds to physical conditions incompatible with any bioactivity.

Since microbes are dynamic systems, both cytoplasm and cell walls undergo continuous changes. This is tantamount to shifting downward or upward both *T*_g,P_ and *T*_g,GP_, or the freezing trends (colored arrows in Figure 1).

In the case of Gram+ bacteria, the *T*_g,PG_ trend strongly depends on the degree of crosslinking between the polymeric components, regulated by specific metabolic activities [34]. This is a reaction against environmental changes [28]. An extreme metabolic reaction is sporulation [48], which implies a severe thickening of the PG shell, whose *T*_g_ can rise up to 90 °C in a P_2_O_5_ dry environment [49]. Taking into account the expected slope of the *T*_g_ trend (see above), if *T*_g,PG_ ≈ 90 °C for *C*_W,PG_ ≈ 0.1 [49], one expects *T*_g,PG_ ≈ −10 °C ≡ 263 K for *C*_W,PG_ ≈ 0.20.

Other metabolic adaptations concern the outer wall of Gram− microbes, where the rigidity/flexibility balance depends on the proportion between saturated and unsaturated lipid moieties of LP and LPS [34].

The lower section of Figure 1 sketches the expected desorption isotherm of a virtual biopolymer, which, for *a*_W_ > 0.85 (where differences between desorption isotherms of large mass polymers become negligible for the scope of the present paper [50,51]), represents the cell polymers at the temperature of the culture. The water activity is the one of the surrounding medium (the only experimentally detectable). This plot allows the identification of the respective critical water activity, *a*_WC,P_ and *a*_WC,PG_. The “no growth *a*_W_” threshold, for *T* = *T*_culture_, coincides with *a*_WC,PG_, as the outer shield would become too rigid for smaller *a*_W_.

It is worth remembering that a 10 K drop of *T*_g_ for an aqueous polymer corresponds to 1% *C*_W_, which implies a very small change in its *a*_WC_. This can explain why the no-growth *a*_W_ of different microbes occurs in the same narrow range of 0.99–1.

The lowest water activity of the medium to allow such residual bioactivity within the cytoplasm is the critical water activity for the virtual polymer that mimics the cytoplasmic macromolecules, *a*_WC,P_ < *a*_WC,PG_. For *a*_W_ < 0.8, no microbe can grow or produce signs of metabolic activity [1,2,4,5].

## 4. The Effect of Temperature

Figure 1 suggests another issue of some relevance. The increase in *T*_culture_ implies lower *a*_WC_ and *C*_WC_, since desorption isotherms do not substantially vary for the range of *a*_W_ of interest. At higher *T*_culture_ (providing that *T*_culture_ < *T*_max_), cellular polymers soften at lower *C*_W_, which means that the entire metabolic machinery can work in drier conditions. The effect of the temperature on the microbial growth, therefore, is a direct consequence of the change in the viscoelastic properties of the cytoplasm and cell wall.

Taking into account that viscosity and elastic modulus depend on *T*, showing an approximately exponential trend, exp(*B*/*T*) (with *B* > 0) [25], one can understand why the growth rate may appear to follow an Arrhenius trend, namely, proportional to exp(−*E*_att_/*T*), in a narrow temperature range [52]. With the aim to improve the estimations of predictive microbiology, Ratkowsky et al. [12] proposed a simple empirical alternative to the misleading Arrhenius equation, namely,(4)$${\left(\sqrt{\mu }\right)}_{Rat}=b\left({T-T}_{min}\right)$$
where *μ* is the maximal specific growth rate, N˙N, *N* being the density of the cell population, and *T*_min_ is the intercept on the *T* axis for *μ* = 0. As openly claimed by the authors [12], “*there is no theoretical foundation for the alternative relationship to be proposed, but it does at least have the virtue of providing an excellent fit to empirical data*”. Based on the above considerations and in line with some suggestions reported in the literature [53], a reasonable physical meaning for *T*_min_ may be*T_min_* = *T*_g,PG_.(5)

According to the growth model proposed in previous works [54,55,56,57], the maximal specific growth rate is(6)$$\mu =ln(2)\frac{3\sqrt{3}}{8}\sqrt{\frac{\beta }{\alpha }}$$
where *α* and *β* are best-fit parameters, which have a biological meaning for the virtual planktonic culture that mimics the overall growth trend of the real culture considered. In particular, *β* represents the theoretical maximum number of duplication steps compatible for the system (intended as medium + cells) and does not depend on the temperature (as long as the volume of the batch culture remains practically unchanged). The parameter *α* instead significantly depends on *T*, α^−1/2^ showing a second-order polynomial trend. This means that $\sqrt{\mu \;}\;\propto \;\alpha^{-\frac{1}{4}}\;\propto T$. Figure 2 shows the corresponding trend.

As for the considered psychrophiles (Figure 2), the intercept for *μ* = 0 corresponds to a subzero temperature, which is the case for many other microbial species [7,12,58]. In the case of the Gram+ *B. subtilis* [49], the values of *T*_min_ are rather close to the expected *T*_g,PG_ (see above). There are no single-cell data about the wall of Gram-negative bacteria, but it seems reasonable to expect a substantially analogous match with the corresponding *T*_min_ values.

With the aim to improve the predictive potential, some authors [59,60,61,62,63,64,65] extended the original Ratkowsky relationship to include two other parameters, namely, *T*_opt_ and *T*_max_, which would correspond to the temperature of the highest *μ* and the maximum temperature compatible with the microbial duplication, respectively. Once again, this approach has no theoretical foundation, as it uses a single function to describe processes of different kinds. The increased number of parameters led to an enhancement of the statistical performance, although at the price of compromising the clear Ratkowsky’s original proposal. The cells of the microbial culture cannot pass from growth to no-growth condition abruptly and simultaneously. This is why one can safely predict a smooth connection between the increasing and decreasing *μ* trends, *T*_max_ reflecting the thermal sustainability of the biochemical processes within the cytoplasm (and exchanges with the surrounding medium) and typically ranging between 40 and 60 °C [58,59,60,61,62,63,64,65] (where the thermally weakest cytoplasmic proteins undergo the unfolding of the native conformation). This produces the appearance of the “fake” maximum at *T*_opt_ (Figure 3).

One could therefore use the *μ* values detected at lower temperatures to determine *T*_min_, according to Equation (4), and tentatively estimate *T*_max_ and, consequently, *T*_opt_, by means of a damping function, *D*(*T*), like the one suggested by Ratkowsky et al. [58], namely,(7)$$D(T)=\left\{1-exp\left[c\left({T-T}_{max}\right)\right]\right\}$$
which reflects the adverse thermal failure of the biochemical machinery. One finally gets the expression(8)$$\sqrt{\mu }={\left(\sqrt{\mu }\right)}_{Rat}D(T)$$
that is zero for *T* = *T*_min_ and *T* = *T*_max_, going through a maximum at *T*_opt_ (Figure 3), although no physical, biochemical, or physiological relationship directly correlates these temperatures to one another. The search for a maximum for √μ reveals that *T*_opt_ depends only on the parameter *c* of the damping function *D*(*T*), as the value of *T*_opt_ is the root of the equation(9)$$exp\left[-c\left({T-T}_{max}\right)\right]=c\left({T-T}_{min}\right)+1$$

The two functions on the left and right sides of Equation (9) are equal to 1 for *T* = *T*_min_ and *T* = *T*_max_, respectively, no matter the value of the parameter *c*, while the related trends cross each other for *T* = *T*_opt_ (Figure 3).

This approach implies reduced predictive performance, but avoids misleading interpretations.

## 5. Conclusions

The above considerations show that the medium water activity and the glass transition temperature of the polymeric components determine the overall behavior of the microbial cells, as their intertwined roles affect every process within the overcrowded intracellular environment and the rigidity of the cell wall. For this reason, *a*_W_ and *T*_g_ are the preferential physical field variables to define the boundaries of the microbial viability and the cell duplication (Figure 4), substantially in line with the view of Pinto and Shimakawa [53], who recognized that the temperature-dependent bacterial growth rate is analogous to that observed in glass-forming liquids in nonliving inorganic materials.

The main conclusion of this work is that, no matter the kind and the physiology (and related biochemistry) of the microbial culture, there are common physical constraints related to the prevailing polymeric nature of the cell components, namely, *a*_WC_ and *T*_g_’, which limit cytoplasmic viability and cell duplication. Beyond these viability boundaries, only spores may exist, although their formation anyway starts within the viability region, and they return to the vegetative state once the viability conditions are restored [66,67]. This conclusion could also explain why both the no-growth *a*_W_ and *T*_min_ have values within similar ranges, namely, 0.99–1 and 260–280 K, respectively, for most microbial species with the exception of some thermophiles [6,58,59,60,61,62,63,64,65].

These limits define the boundaries of an aqueous liquid phase, the only one where molecular mobility can sustain the progress of the biochemical processes and the related irreversible production of entropy [68].

## Figures and Tables

**Figure 1 ijms-26-08508-f001:**
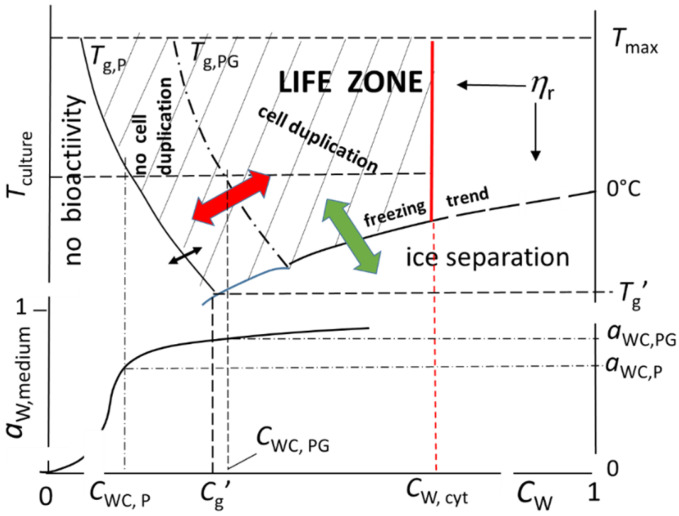
Upper section: Dynamic phase diagram that qualitatively reflects the properties of the aqueous fractions of a microbial cell. *C*_W_ is the water mass ratio. The region with *C*_W_ > *C*_g_’ corresponds to a condition of low viscosity, just as in any aqueous solution of SMMS. The *C*_W_ < *C*_g_’ region reflects the viscoelastic behavior of the hydrated macromolecular components, namely, polymers of cytoplasm and external envelope, which are represented by two virtual polymers that mimic their glass transition trends, *T*_g,P_ and *T*_g,GP_, respectively. *T*_max_ is the highest temperature compatible with cell survival. Lower section: Desorption isotherm (at the temperature of the microbial culture) of the hydrated polymers P and PG, where *a*_WC,P_ and *a*_WC,PG_ are the corresponding critical water activity.

**Figure 2 ijms-26-08508-f002:**
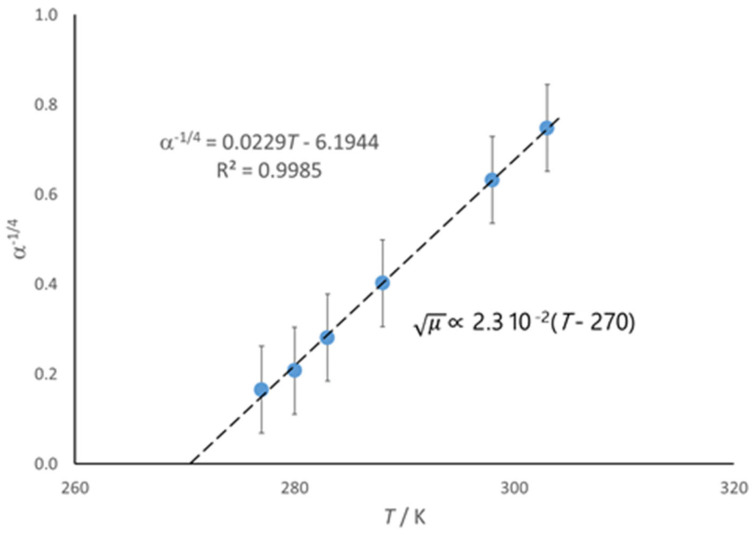
The square root of the maximal specific growth rate, *μ*, is proportional to *α*^−1/4^ (see text), which shows a straight-line trend with an intercept at 270 K. Average values of the parameter *α* from the fit of growth curves of psychrophiles *Aeromonas hydrophila*, *Listeria monocytogenes*, and *Yersinia enterocolitica* at various temperatures [57].

**Figure 3 ijms-26-08508-f003:**
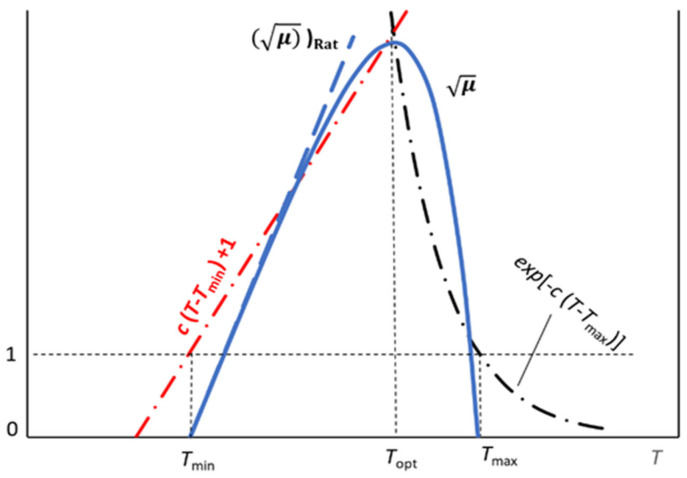
The trend of the square root of the maximal specific growth rate, *μ* (continuous bell-shaped blue line), goes through a maximum at *T* = *T*_opt_ that satisfies Equation (9). The dotted blue straight line corresponds to the original Ratkowsky empirical trend. *T*_min_ should coincide with the glass transition temperature of the outer shell, *T*_g,PG_ (see text), while *T*_max_ is the highest temperature tolerated by the cytoplasmic biochemical machinery.

**Figure 4 ijms-26-08508-f004:**
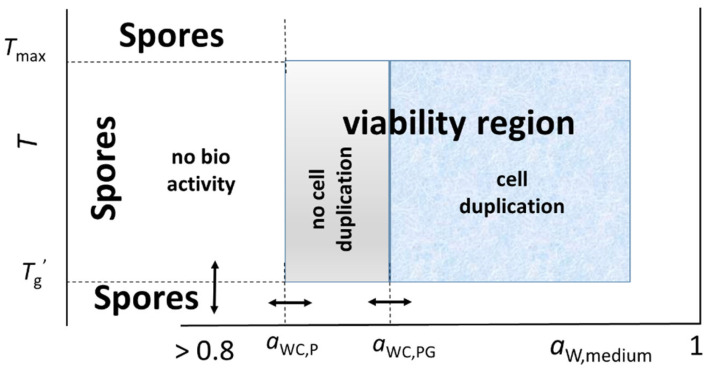
Water activity and glass transition temperature of the polymeric cytoplasmic components and outer envelope are the field variables that define the favorable conditions for the viability and duplication of microbial cells within a narrow “viability region”. *T*_g_’ is the lowest freeze temperature (see Figure 2). Outside this region, spores are the only surviving form of life.

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
