# Peer review of "Microbial Growth: Role of Water Activity and Viscoelasticity of the Cell Compartments"

_ijms, 2025, doi:10.3390/ijms26178508_

Round 1

Reviewer 1 Report

Comments and Suggestions for Authors

This paper aims to explore the regulatory role of water activity (aW​) and viscoelasticity of cell compartments in microbial growth.

Line 23: In the abstract, “ single paragraph of about 23 200 words maximum. For research articles, abstracts should give a pertinent overview of the work. We strongly encourage authors to use the following style of structured abstracts, but without headings: (1) Background: Place the question addressed in a broad context and highlight the purpose of the study; (2) Methods: briefly describe the main methods or treatments applied; (3) Results: summarize the article’s main findings; (4) Conclusions: indicate the main conclusions or interpretations. The abstract should be an objective representation of the article and it must not contain results that are not presented and substantiated in the main text and should not exaggerate the main conclusions.” This suggests that the abstract of this article was likely generated by AI and therefore does not comply with academic standards.

The introduction section has significant deficiencies in elaborating the research background:

There is a lack of systematic review on the relationship between "water activity–glass transition–microbial growth." The logical connection between existing studies (e.g., classic works by Labuza, Rahman, etc., on a W and microbial stability in food systems) and the hypotheses of this paper is not sufficiently,leading to an unclear scientific rationale for the proposed research questions.

References to core concepts such as "cytoplasmic crowding" and "membrane-less phase separation" (e.g., Ellis 2001, Parry 2014) are merely listed without in-depth analysis of their direct relevance to microbial growth, making the innovation and necessity of the research objectives ambiguous.

Methodology: As an interdisciplinary study involving thermodynamics, rheology, and microbiology, the paper fails to clearly elaborate on key experimental methods and data sources, directly affecting the credibility of the results.

The core hypothesis of this paper (that water activity and glass transition regulate microbial growth) has certain potential value. However, due to the lack of methodology, insufficient experimental evidence, and broken logical chains, it cannot support the scientificity and reliability of its conclusions. It is recommended that the authors supplement key experimental data (e.g., correlation between Tg​ and growth curves of multiple species, dynamic measurement of aW​ and cell viscoelasticity), clarify the physical meaning of model parameters, and reorganize the logical structure before resubmission.

Author Response

I thank you for the comments and recommendations. The manuscript is now thoroughly changed.

  1. The abstract now should comply with your suggestion. I have anyway to add that, because of my age, I am not familiar at all with AI and confirm that I did not use such an approach to prepare this article (and all my previous almost 200 publications).
  2. This thoroughly revised version explicitly reports that the main proposal of the paper is to extend the polymer science approach (widely used in food technology) to the microbial cultures. This approach gives relevance to the viscoeleastic properties of the systems considered, with particular emphasis to the molecular mobility, which, also in the case of microbial cultures, governs every process.
  3. The discussion on the cytoplasmic crowding is now limited to the misinterpreted “osmotic” effect of the macromolecules, which is a serious and unfortunately widely spread mistake (in the eyes of a thermodynamicist) that led to the wrong proposal of a revaluation of the association constants. These were indeed very correctly determined at as low as possible concentration. What needs a revaluation is the kinetics of the association, which directly depends on the concentration (and not on the thermodynamic activities) and is a diffusion-limited process. This point is explicitly, although softly, mentioned in the paper.
  4. I added a number of references related to the experimental approaches, discussing their potential support to the polymer science approach proposed in the paper, with specific reference to the need of data related to the properties of the single cells, rather than to the overall viscoelastic properties of the microbial culture.

Reviewer 2 Report

Comments and Suggestions for Authors

The present paper critically reviews thermodynamic and rheological approach that defines common physical constraints that govern the overall behavior of batch planktonic microbial cultures. The manuscript can be considered for publication after addressing the comments given below.

My detailed suggestions are as follows:

  1. Manuscript lacks summary and refinement, and needs overall improvement.
  2. Abstract must have rationale, objective, materials and methods and conclusions. First sentence must be a rationale. Please re-write abstract and delete: “For research articles, abstracts should give a pertinent overview of the work. We strongly encourage authors to use the following style of structured abstracts, but without headings: (1) Background: Place the question addressed in a broad context and highlight the purpose of the study; (2) Methods: briefly describe the main methods or treatments applied; (3) Results: summarize the article’s main findings; (4) Conclusions: indicate the main conclusions or interpretations. The abstract should be an objective representation of the article and it must not contain results that are not presented and sub stantiated in the main text and should not exaggerate the main conclusions.”
  3. The abstract and Introduction sections also lacks clarity regarding the novelty of the work.
  4. I did not see the purpose of this paper. Methodologically there is no review. Review title should introduce the object and subject of research.
  5. The Author should note in the title the subject of their research and expand this into the purpose of the paper, which is not there yet.
  6. Section 2 is too long and lacks a sense of hierarchy. Please correct it. The writing is neither concise nor precise, and the paragraph structure lacks logical coherence.
  7. The title, purpose, and conclusion should be derived from each other. The Author should make their review stand out in some way.
  8. Please supplement the mechanism depending on the cell wall of Gram-positive bacteria and of Gram-negative bacteria.
  9. The review should be more on compilation of discussion and figures.
  10. Please provide the research prospects for it possible application in practice.

Author Response

Response to Reviewer 2.

I thank you for the comments and recommendations. The manuscript is now thoroughly changed.

  1. The abstract now should comply with your suggestion.
  2. This thoroughly revised version explicitly reports that the main proposal of the paper is to extend the polymer science approach (widely used in food technology) to the microbial cultures. This approach gives relevance to the viscoeleastic properties of the systems considered, with particular emphasis to the molecular mobility, which, also in the case of microbial cultures, governs every process.
  3. The new version has a much more concise section 2 (the old figure 1 was canceled) , trying to show why it is necessary to briefly review the properties of the cytoplasm in view of explain how the dynamic state diagram can account for the consequence of the reduced molecular mobility within a crowded environment..
  4. I have extended the discussion to the case of Gram- bacteria and their peculiar cell envelope.
  5. The presentation now reports many references related to the experimental approaches, discussing their potential support to the polymer science approach proposed in the paper, with specific reference to the need of data related to the properties of the single cells, rather than to the overall viscoelastic properties of the microbial culture.

Round 2

Reviewer 1 Report

Comments and Suggestions for Authors

The article, titled "Microbial Growth: The Role of Water Activity and Viscoelasticity of Cellular Compartments," reexamines the physical limits of microbial growth from a polymer science perspective. It proposes a framework based on water activity (WA) and glass transition temperature (TgTG) to explain the boundaries of microbial growth under different environmental conditions. The authors skillfully apply theories from food and materials science to microbiology, highlighting the critical role of the viscoelastic properties of the cytoplasm and cell envelope in cell survival and replication, while also providing a potential physical explanation for the Ratkowski equation.

While this article is a perspective piece without new experimental data, its theoretical framework is sound and well-supported by the literature. It has significant academic value in complementing and refining existing microbial growth models. Particularly compelling are the new insights into the explanation of the intracellular-extracellular water activity imbalance, the distinction between the glassy behavior of the cell wall and cytoplasm, and the physical interpretation of the Ratkowski equation.

In summary, after revision, this manuscript has reached the publication standard in terms of theoretical depth, logical rigor, and clarity of expression. We recommend acceptance.

Reviewer 2 Report

Comments and Suggestions for Authors

The authors have made good improvements to the manuscript and addressed all the queries.